# Perturbations in the Heme and Siroheme Biosynthesis Pathways Causing Accumulation of Fluorescent Free Base Porphyrins and Auxotrophy in *Ogataea* Yeasts

**DOI:** 10.3390/jof7100884

**Published:** 2021-10-19

**Authors:** Azamat V. Karginov, Alexander I. Alexandrov, Vitaly V. Kushnirov, Michael O. Agaphonov

**Affiliations:** Federal Research Center “Fundamentals of Biotechnology” of the Russian Academy of Sciences, 119071 Moscow, Russia; karginovaz@mail.ru (A.V.K.); alexvir@gmail.com (A.I.A.); vkushnirov@mail.ru (V.V.K.)

**Keywords:** porphyrin, siroheme, CRISPR-Cas9, yeast transformation, auxotrophy, selectable marker, non-conventional yeasts

## Abstract

The biosynthesis of cyclic tetrapyrrol chromophores such as heme, siroheme, and chlorophyll involves the formation of fluorescent porphyrin precursors or compounds, which become fluorescent after oxidation. To identify *Ogataea polymorpha* mutations affecting the final steps of heme or siroheme biosynthesis, we performed a search for clones with fluorescence characteristic of free base porphyrins. One of the obtained mutants was defective in the gene encoding a homologue of *Saccharomyces cerevisiae* Met8 responsible for the last two steps of siroheme synthesis. Same as the originally obtained mutation, the targeted inactivation of this gene in *O. polymorpha* and *O. parapolymorpha* led to increased porphyrin fluorescence and methionine auxotrophy. These features allow the easy isolation of Met8-defective mutants and can potentially be used to construct auxotrophic strains in various yeast species. Besides *MET8*, this approach also identified the *HEM3* gene encoding porphobilinogen deaminase, whose increased dosage led to free base porphyrin accumulation.

## 1. Introduction

Tetrapyrrole compounds such as heme, chlorophyll, cobalamin, etc., accomplish many essential functions in different organisms (for a review see [1]). Their synthesis begins with formation of aminolevulinic acid either from glutamine (in plants, archaea, and the vast majority of bacteria) or from glycine and succinyl-CoA (in animals, fungi, and the α-subclass of the photosynthetic purple bacteria) [2]. Aminolevulinic acid is used to form porphobilinogen, four molecules of which are then condensed to form a linear tetrapyrrole, uroporphyrinogen I. The latter compound is converted into a cyclic tetrapyrrole, uroporphyrinogen III. This precursor is sequentially converted into coproporphyrinogen III, protoporphyrinogen IX, and protoporphyrin IX to form heme and, in case of plants and algae, chlorophyll. Defects in different steps of heme synthesis in humans may lead to disorders, which are called porphyrias and are accompanied with accumulation of fluorescent porphyrin compounds (for a review see [3]). Among intermediate compounds in this pathway only protoporphyrin IX manifests bright fluorescence peaking at ~630 nm, which is excited by ~400 nm light [4]. The incorporation of Fe ion into protoporphyrin IX during heme formation leads to the loss of this fluorescence. At the same time, some other intermediates can be non-enzymatically oxidized to form fluorescent porphyrins such as uroporphyrin and coproporphyrin [3].

An intermediate compound in heme biosynthesis, uroporphyrinogen III, is also used in a branch of this pathway where it is modified in three steps to form siroheme. This compound is the prosthetic group required for the six-electron reduction of both sulfite and nitrite. The first step of conversion of uroporphyrinogen III into siroheme is its methylation, producing precorrin-2. In yeast, this step is catalyzed by Met1, while dehydrogenase and chelatase activities of Met8 are required for the remaining two steps, namely sirohydrochlorin formation and the incorporation of Fe^2+^ [5]. The inactivation of either of these proteins leads to methionine auxotrophy due to the inability to utilize sulfur in a form of sulfate or sulfite.

*Ogataea polymorpha* and *O. parapolymorpha* are very closely related methylotrophic thermotolerant yeast species, which are frequently used as hosts for recombinant protein production [6]. Besides this application, these yeasts are used as a model organisms in studies of methanol assimilation, peroxisome biogenesis and destruction [7], protein secretion and glycosylation [8], nitrite and nitrate assimilation [9], sugar metabolism [10], heat tolerance [11], and cell division [12]. However, the tetrapyrrole synthesis pathways in this yeast remain uncharacterized. Here, we demonstrate that, in these yeasts, screening for colonies emitting red fluorescence upon irradiation with ~400 nm light allows for the selection of clones with alterations in heme and siroheme biosynthesis pathways. Specifically, by using this approach, we isolated *O. polymorpha* and *O. parapolymorpha* mutants lacking dehydrogenase/chelatase Met8, which is responsible for siroheme formation. Since this mutation leads to methionine auxotrophy, the *MET8* gene can be used as a selectable marker. In addition, we found that overproduction of porphobilinogen deaminase causes free base porphyrin accumulation.

## 2. Materials and Methods

### 2.1. Yeast Strains and Media

Rich YP medium (1% yeast extract, 2% peptone) supplemented either with 2% glucose (YPD) or 1% glycerol (YP-Glycelrol), as well as synthetic medium SC-D (0.67% Yeast Nitrogen Base, 2% glucose) were used to cultivate yeast strains.

The *leu2* auxotrophic mutants of *O. polymorpha* and *O. parapolymorpha* DL-1, A16 [13] and DL1-L [14], respectively, as well as *O. polymorpha* 1B (*leu2 ade2*) [15] and 1B27 (*leu2 ade2 ura3::ADE2*) [16] double auxotrophic strains were used in this study. The strains mentioned above DL1-L, A16 and 1B were provided by Dr. M. Beburov (Research Institute of Genetics and Selection of Industrial Microorganisms, Moscow, Russia), while 1B27 was created by the replacement of a portion of the *URA3* gene for *ADE2* in the 1B strain. The derivative of A16, A16-SSAMC strain, whose *SSA1* open reading frame (ORF) was fused to sequence coding for the mCherry, was used for the selection of mutants with enhanced red fluorescence during irradiation by 400 nm light. Integration of mCherry coding sequence into A16 genome was carried out as it was described previously for tagGFP [17]. The Ssa1-mCherry fusion exhibited very poor fluorescence that was originally intended to be used for selection mutants with increased mCherry fluorescence, but this was not the aim of the present study.

### 2.2. Yeast Transformation

Yeasts were transformed using the Li acetate method [18] with some modifications as follows. Cells from 300 μL of an exponentially grown culture were harvested by centrifugation in a bench-top microcentrifuge at 5000 rpm for 30 s, washed, and re-suspended in 42 μL of sterile water. Then, 2 μL of the DNA-carrier solution (10 mg/mL, shared and denatured by boiling) and 6 drops (approx. 90 μL) of 70% PEG 4000 were added and mixed well. After that, the suspension was mixed with 9 μL of 1 M Li-acetate solution and dispensed by 20–23 μL to add 1 μL of transforming DNA to each portion. Suspensions were incubated first at 30 °C for 30 min than at 45 °C for 30 min. The cells were washed with YPD medium and spread on plates for the selection of transformants.

### 2.3. Plasmids and Genomic Library Construction

The construction of *O. polymorpha* genomic library was described previously [19]. The *met8* deletion allele was obtained by removing first 75 codons of *MET8* ORF. To do this, the 4654 bp *Hin*dIII-*Ecl*136II fragment of the plasmid pAM898-10, which was isolated from the genomic library by complementation of the P6 mutant auxotrophy, was sub-cloned between *Hin*dIII-*Ecl*136II sites of the pTZ18 vector with preliminary removed *Eco*RI site. Then, the *Bsp*1407-*Eco*RI fragment was deleted in the resulting plasmid. The plasmid pWS173-LEU2-Amp possessing the guide RNA assembly cassette and Cas9 encoding gene was constructed by subcloning of the 7 kb pWS173 [20] Eco72-SpeI fragment between the Ecl136II and XbaI sites of YEpLac181 [21]. To obtain the gene of guide RNA targeting Cas9 cut to the *MET8* sequence, which was removed in the constructed disruption allele, this plasmid was digested with Esp3I and ligated with the duplex of oligonucleotides GACTAGCAGATGCACAGATCACAG and AAACCTGTGATCTGTGCATCTGCT. The resulting plasmid was designated as pAM949.

To achieve the multiple integration of the AMIpSL1-based plasmids into the genome, the previously described stabilization procedure was applied to transformants bearing such plasmid in autonomous state [22]. In particular, freshly obtained transformants were streaked on selective medium plates to obtain single colonies. The largest colonies were picked up and steaked again. This procedure was repeated until all subclones appeared after streaking show the same growth rate. As a rule, such subclones possess 5–10 copies of the plasmid integrated into a telomere region as tandem repeats.

### 2.4. Selection of Mutants Accumulating Free Base Porphyrins

The A16-SSAMC strain was mutagenized as follows. Cells from overnight YPD culture were collected by centrifugation and resuspended in sterile water at OD_600_ = ~0.5. The suspension was irradiated by a mercury-vapor quartz lamp for 20 s. Then cells were suspended in YPD, incubated at 37 °C for 2 h and spread on YPD plates in aliquots producing ~500 colonies per plate. To detect fluorescent colonies, they were irradiated by a 1 W light emitting diode (LED) with λ_peak_ = 402 nm and Δλ_1/2_ = 15 nm and viewed through a red filter in order to enhance the signal to noise ratio. The LED emission wavelength was measured by a portable MK350S Premium spectrometer (UPRtek, Zhunan, Taiwan). The LED was equipped with a narrow angle collimator lens to increase light brightness.

### 2.5. Analysis of Free Base Porphyrins Accumulated in Yeast Cells

To extract water soluble porphyrins, cells from 5 mL YPD overnight culture were collected by centrifugation, washed with distilled water, and mixed with 1 mL of chloroform by vortexing. The mixture was shaken for 1 h and centrifuged at 15,000× *g* for 5 min. The upper phase was transferred to a new tube. The lower phase was mixed with 150 μL of distilled water, vortexed for several minutes, and centrifuged again for phase separation. The upper phases were joined and used for spectrofluorimetry analysis. The pH of the resulting extract was approximately 5.0–5.5. To perform extraction with 50% methanol, cells were disrupted by vortexing with glass beads in 20 mM Tris-HCl buffer pH 8.0 supplemented with 2 mM EDTA. The obtained cell homogenate was mixed with equal volume of methanol, vortexed for 5 min and centrifuged at 15,000× *g* to remove precipitate and cell debris. To perform extraction with virtually 100% methanol, cells were collected by centrifugation, washed with distilled water, and mixed with 1 mL of chloroform by vortexing. The mixture was shaken for 30 min and centrifuged at 15,000× *g* for 5 min. The upper (water soluble) and lower (chloroform) phases were removed to retain interphase, which was then mixed with 250 μL of methanol. The obtained mixture was shaken for 1 h and then centrifuged to remove insoluble matter. Spectra were acquired using a FluoroMax-4 spectrometer (HORIBA Scientific).

Flow cytometric analysis was performed using the CytoFLEX S flow cytometer (Beckman Coulter) with 405 nm excitation laser and 610/20 nm bandpass filter. A total of 10,000 cells were acquired in each sample.

## 3. Results

### 3.1. Inactivation of the MET8 Gene in O. polymorpha and O. parapolymorpha Confers Red Fluorescence Excited by ~400 nm Light

To identify mutations affecting synthesis of tetrapyrrole compounds in the methylotrophic yeast *O. polymorpha*, we performed a screen for mutants accumulating free base porphyrins. Such mutants were selected in the strain A16-SSAMC (see Materials and Methods) after UV mutagenesis. The selection was carried out by searching for colonies emitting red fluorescence when irradiated by a 402 nm light emitting diode (LED). Approximately 0.2% colonies obtained after mutagenesis emitted visible fluorescence. Five clones with the brightest fluorescence designated P3, P6, P12, P16, and P25 were selected for further experiments. The P6 mutant, in addition to the red fluorescence, manifested an inability to grow on SC-D medium, indicating some auxotrophy. This phenotype was used for cloning the wild-type allele of the mutated gene by complementation. Several clones able to grow on SC-D medium were obtained by transformation with an *O. polymorpha* genomic library. Plasmids from three of them were isolated via *E. coli* transformation and sequenced. In all cases, the plasmids contained inserts with an open reading frame (ORF), corresponding to locus 1018755–1019542 bp of *O. polymorpha* chromosome 4 (GenBank accession number CP060325), encoding a homologue of *S. cerevisiae* Met8, which is responsible for the last steps of siroheme synthesis. The alignment of protein sequences revealed a conservative region possessing the aspartyl residue, which was previously shown [23] to be essential for both the dehydrogenase and ferrochelatase activity of the Met8 protein (Figure 1).

We designated the cloned gene as its *S. cerevisiae* homologue *MET8*. *O. polymorpha MET8* ORF was interrupted by a 32 bp intron at the 21st codon. Further analysis showed that the inability of the P6 mutant to grow on SC-D was due to methionine auxotrophy. Notably, the *S. cerevisiae met8-*Δ mutant from the Euroscarf collection did not display increased red fluorescence upon irradiation with the 402 nm LED. To prove that the inactivation of *MET8* in *O. polymorpha* and *O. parapolymorpha* leads to increased fluorescence excited by 402 nm LED, these yeasts were transformed with a fragment of this locus possessing a deletion within the *MET8* gene (see Materials and Methods). In case of *O. polymorpha* DL1-L strain, the transformation mix was spread onto complex YPD medium to obtain single colonies without any additional selection. Only one clone with red fluorescence was revealed among several thousand colonies obtained from the transformation mix (Figure 2). This clone was a methionine auxotroph that indicated the inactivation of the *MET8* gene.

According to our experience, frequency of DNA fragment integration via homologous recombination in *O. parapolymorpha* is several folds higher than in *O. polymorpha*. To increase the probability of obtaining *MET8* disruptants in *O. polymorpha* A16, this strain was co-transformed with a *MET8* disruption allele and an autonomously replicating plasmid, which could be lost from obtained transformants. Since this plasmid carried *LEU2* gene as a selectable marker, the transformants could be selected on leucine omission medium getting rid of the cells, which were not competent for transformation. As a result, we could select one Met^−^ colony by monitoring red fluorescence, though the difference in the fluorescence intensity between this and other transformants on the plate appeared to be quite modest. This was apparently due to a negative effect of the medium composition on porphyrin accumulation in the *met8-*Δ mutant, since cells grown on synthetic SC-D medium, which was used for the selection of the transformants, exhibited much less fluorescence than cells grown in rich YPD medium (Figure 3). The presence of the deletion in the *MET8* locus in the obtained clones was confirmed by PCR analysis.

One could expect that an improvement in frequency of obtaining of *met8* mutants could be achieved using the CRISPR-Cas9 system. To do this in *O. polymorpha*, we applied a plasmid with the guide RNA assembly cassette and Cas9 encoding gene, which was originally designed for *S. cerevisiae* (see Materials and Methods). A plasmid (pAM949) was created in order to direct Cas9-mediated cleavage of the *MET8* gene and was subsequently used to transform the *O. polymorpha* 1B27 strain. Transformation efficiency with this plasmid was quite low. Possibly this could be attributed to the large size of the plasmid (~12 kb) or to its poor ability to be maintained autonomously in *O. polymorpha* cells. We expected that the double strand break produced by Cas9 would be repaired by the non-homologous end joining process, which may result in mutations affecting Met8 expression or function. Indeed, among ~20 Leu^+^ transformants, obtained after transformation of the *O. polymorpha* 1B27 strain with this plasmid, four had different levels of red fluorescence. One such clone exhibiting the brightest fluorescence was a methionine auxotroph. The remaining fluorescent clones probably possessed *MET8* mutations, which only partially reduced function of the corresponding protein.

To induce deletion in *MET8* using CRISPR-Cas9, the *O. polymorpha* 1B strain was transformed with a mixture of the pAM949 plasmid (coding for Cas9 and guide RNA) and DNA fragment representing a *MET8* deletion allele. Among several clones obtained after transformation, one methionine auxotroph was identified. According to PCR analysis it possessed the *MET8* deletion allele, which emerged due to CRISPR-Cas9-induced recombination of the transforming DNA fragment with the chromosomal *MET8* locus.

### 3.2. An Increase in the HEM3 Gene Dosage Causes Accumulation of Free Base Porphyrins

We also attempted to clone wild-type alleles of genes, whose mutations led to porphyrin accumulation in other mutants, which were not auxotrophs. This was performed by transforming clones P3, P12, P16, and P25 with an *O. polymorpha* genomic library and searching for clones with reduced fluorescence. However, no such clones were revealed. At the same time, clones with brighter fluorescence were revealed among transformants of the P12 mutant. The plasmid isolated from one such clone carried a gene (GenBank accession number BBNV01000003, region 697510–698872 bp, lower strand), coding for a homologue of porphobilinogen deaminase. This gene also possessed an intron interrupting its ORF at the 184th codon. This appears to be an unusual intron position, since in yeast, introns are usually located near the ORF start. The identified gene was the only gene coding for a homolog of this enzyme, indicating that it has the same function. In *S. cerevisiae,* this enzyme is encoded by the *HEM3* gene, and we also designated the identified gene as *HEM3*. Compared to the *S. cerevisiae* protein, *O. polymorpha* Hem3 has an ~80 amino acid residue N-terminal extension, which is also present in the *Candida boidinii* homolog (Figure 1).

Then, we asked whether the ability of extra copies of the *HEM3* gene to increase the accumulation of free base porphyrins is specific to the P12 mutant, in which this gene was cloned. To address this question we transformed other mutants, namely P16 and P3, as well as the wild-type strain with the plasmid bearing *HEM3*. This plasmid was based on the AMIpSL1 vector, which allows the selection of clones resulting from multiple integration of the plasmid into the genome [22]. These clones were subjected to the stabilization procedure applied to the obtained transformants (see Materials and Methods). All of them manifested brighter fluorescence compared to the strains with the empty vector. This proved that the effect of the *HEM3* gene was not specific to the P12 mutant.

### 3.3. Fluorescence Spectra Reveal Distinct Types of Free Base Porphyrins Accumulated in Different Mutants

Although the clones accumulating free base porphyrins were identified by red fluorescence in response to irradiation with a 402 nm LED, one can expect that different mutations should lead to accumulation of different fluorescent compounds. To study this, we extracted the fluorescent compound(s) from the identified mutants and acquired fluorescence spectra. First, we attempted to extract the fluorescent compound from the *met8-*Δ mutants by the treatment of cells with chloroform and methanol [24] according the protocol described in [25]. However, this method provided very poor extraction probably due to low solubility of the fluorescent compound in the methanol-water mix. To improve the extraction, we excluded methanol from the procedure (see Materials and Methods). The fluorescence was detected only in the water-soluble phase, though it likely contained numerous impurities, which interfered with the detection of porphyrin excitation bands other than the most intense short-wave Soret band (Figure 4).

The same extraction procedure was also efficient in the mutants P3, P12, and P16, which allowed us to acquire the fluorescence spectra of their free base porphyrins in the water-soluble fraction (Table 1). Unlike the *met8* mutant extract, whose emission spectrum possessed only one pronounced peak around 600 nm, the spectra of the P12 and P16 mutants had two emission peaks at this wave-length region and 680 nm peak, which is probably the second peak of one of the porphyrins producing the shorter wavelength peaks. The spectrum of water-soluble extract of the P3 mutant had 4 peaks in the region around 600 nm. Multiple emission peaks indicated the presence of different porphyrin compounds in these mutants. The spectra of the P12 and P16 mutants were very similar evoking the suggestion that the same gene was altered in these mutants.

The extraction procedure, which was used for mutants described above, was not effective in P25 mutant since the fluorescent matter remained in the interphase together with the cell debris, indicating that this compound is poorly soluble in both chloroform and water phases, at least at pH around 5. This was overcome when cells homogenates obtained by vortexing with glass beads were mixed with methanol to achieve a final alcohol concentration around 50%. This released the fluorescent compound into the soluble fraction that allowed us to acquire fluorescence spectra (Table 1). Extraction from the water-chloroform interphase with pure methanol (see Materials and Methods) was less efficient, but also allowed acquisition of fluorescent spectra. The major emission peak was at 626 or 629 nm, which was followed by a smaller peak at 685 or 690 nm in case of 50% and pure methanol extractions, respectively. This resembles spectra of protoporphyrin IX acquired in water and pure methanol solutions [4]. Moreover, protoporphyrin IX is poorly soluble in water and was shown to aggregate at neutral pH [26].

The 50% and pure methanol extraction procedures were also efficient in mutants P3, P12 and P16; however, the number of fluorescence emission bands differed from that in the water-soluble extracts. Weak porphyrin fluorescence was also revealed in the wild-type strain sample obtained by the pure methanol extraction (Table 1).

Although we succeeded in identification of a gene whose defect causes fluorescent porphyrin accumulation only in one mutant, the differences in the properties of porphyrins revealed in other mutants indicate that most of these mutants (except P12 and P16) have different alterations in porphyrin biosynthesis pathways.

## 4. Discussion

Here, we isolated *O. polymorpha* mutants accumulating free base porphyrins and characterized two *O. polymorpha* genes involved in tetrapyrrole biosynthesis. One of them, designated as *MET8*, codes for ortholog of the bifunctional dehydrogenase and ferrochelatase, which catalyzes oxidation of precorrin-2 to sirohydrochlorin and incorporation of Fe ion to form siroheme [5]. The latter reaction abolishes porphyrin fluorescence at ~600 nm excited by 400 nm light. One can expect that the loss of Met8 should lead to accumulation of precorrin-2, which does not exhibit porphyrin-specific fluorescence. Possibly, precorrin-2 is oxidized to a fluorescent compound(s) independently of Met8. On the other hand, the accumulation of precorrin-2 may in turn lead to the accumulation of uroporphyrinogen III, which is not a fluorescent compound either but can be non-enzymatically oxidized to fluorescent uroporphyrin. Although we have not identified the fluorescent compound, our results demonstrate that *met8* auxotrophic mutants can be selected in some yeast species by searching for red fluorescent clones upon irradiation with ~400 nm light. This can be a useful tool for the genetic manipulation of strains lacking auxotrophic selectable markers or when introducing an additional selectable marker is required.

Another gene designated as its *S. cerevisiae* homolog *HEM3*, which encodes porphobilinogen deaminase, was identified due to its ability to induce the accumulation of free base porphyrins in multi copy state. In *S. cerevisiae*, this gene is not regulated by heme level [27] which may explain the pronounced effect of its increased dosage in *O. polymorpha*. Indeed, the absence of negative feedback regulation should allow the increased gene dosage to result in the increased production of porphobilinogen deaminase. This should in turn increase the synthesis of uroporphyrinogen I and possibly some other tetrapyrrole intermediates, which can be converted into fluorescent porphyrins.

Thus, our results demonstrate that monitoring the porphyrin-specific fluorescence allows for a selection of clones with perturbed heme or siroheme biosynthesis pathways in yeast *O. polymorpha* and *O. parapolymorpha*.

## Figures and Tables

**Figure 1 jof-07-00884-f001:**
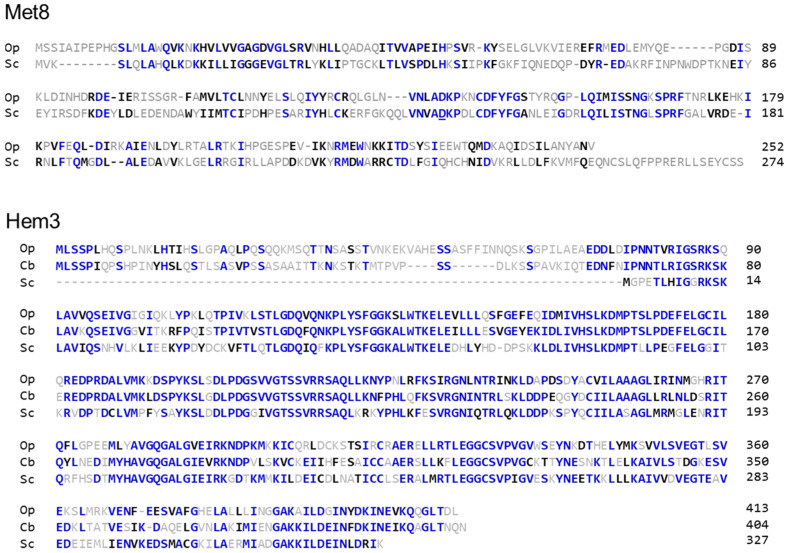
Alignments of deduced protein sequences. Met8, alignment of protein sequences encoded by *O. polymorpha* (Op) and *S. cerevisiae MET8* genes. Hem3, alignment of porphobilinogen deaminase sequences of *O. polymorpha* (Op), *Candida boidinii* (Cb), and *S. cerevisiae* (Sc). Similar residues are shown in black; identical residues are shown in blue. The aspartate residue, which is essential for both dehydrogenase and ferrochelatase activity of the Met8 protein is underlined.

**Figure 2 jof-07-00884-f002:**
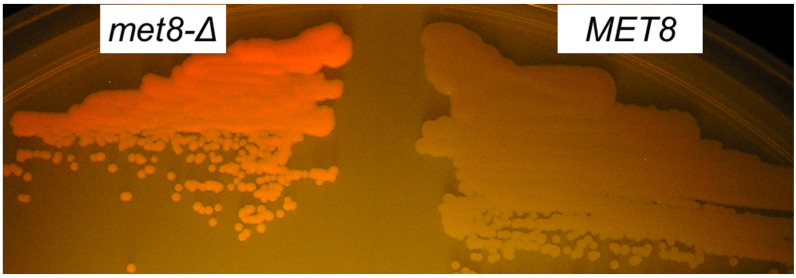
Fluorescence of *O. parapolymorpha met8-*Δ colonies irradiated by 402 nm LED light. The *met8-*Δ mutant (*met8-*Δ) and original strain with wild-type *MET8* allele (*MET8*) were streaked on YP-1% Glycerol medium and incubated for 2 days to obtain single colonies, which were irradiated by 402 nm LED and photographed through an orange filter.

**Figure 3 jof-07-00884-f003:**
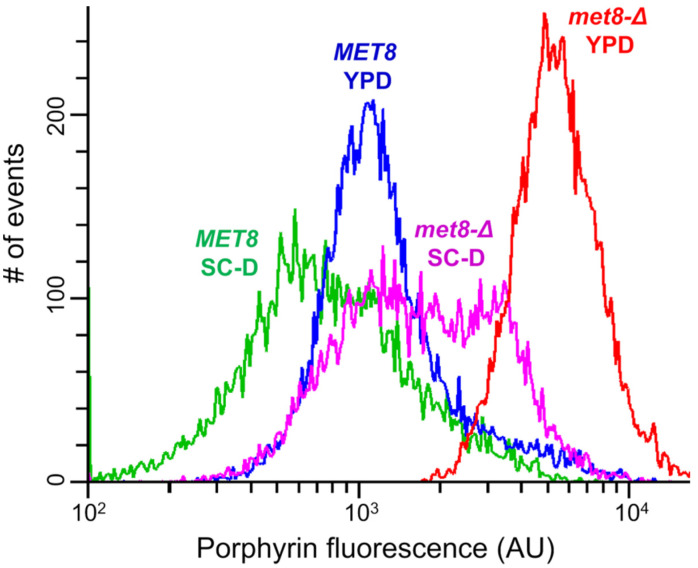
Distribution of cells according to porphyrin fluorescence in cultures of *O. parapolymorpha* strains with wild-type (*MET8*) and deletion (*met8-*Δ) *MET8* alleles, which were grown in the synthetic SC-D or complex YPD medium.

**Figure 4 jof-07-00884-f004:**
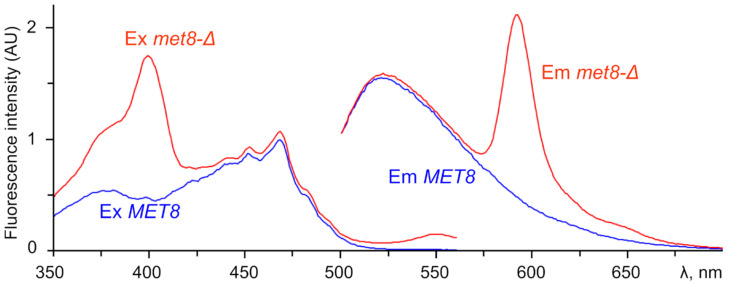
Excitation (Ex) and emission (Em) spectra (recorded at λ_em_ = 595 nm and λ_ex_ = 400 nm, respectively, with 5 nm bandwidth and 1 nm resolution) of cell extracts of *O. parapolymorpha met8-*Δ mutant (red) and strain with *MET8* wild-type allele (blue).

**Table 1 jof-07-00884-t001:** Fluorescence emission and corresponding Soret excitation bands (nm) in cell extracts of mutants accumulating free base porphyrins.

Mutant	Extraction Solvent	Emission	Excitation
*met8-*Δ	H_2_O	594	399
P3	H_2_O	570	412
		593	411
		606	401
		641	410
	50% Methanol	578	410
		620	399
		650	399
		683	399
	Methanol	621	398
		686	399
P12	H_2_O	580	411
		619	400
		680	398
	50% Methanol	577	410
		620	400
		650	399
		684	398
	Methanol	621	399
		685	399
P16	H_2_O	580	411
		619	400
		680	398
	50% Methanol	579	410
		621	399
		650	398
		680	399
	Methanol	621	399
		684	399
P25	50% Methanol	626	399
		685	399
	Methanol	629	400
		690	400
W. T.	Methanol	619	399

## Data Availability

Not applicable.

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
