# Peer review of "Perturbations in the Heme and Siroheme Biosynthesis Pathways Causing Accumulation of Fluorescent Free Base Porphyrins and Auxotrophy in *Ogataea* Yeasts"

_jof, 2021, doi:10.3390/jof7100884_

Round 1

Reviewer 1 Report

The submitted manuscript represents novel, interesting and important work, especially for readers working in the same field. The overall merit of the manuscript is, in my opinion, enough to be accepted and published in Journal of Fungi. 

The only aspect that, in my opinion, could be improved is the english writing, especially in the beginning of sentences and paragraphs, but it is not reason enough the reject or recommend revisions in the manuscript. In this way, I recommend acceptance of the manuscript in the current form, and authors could try to improve the writing in the final revision round.

Author Response

The manuscript has been edited by a native English speaker with a proper scientific background.

Reviewer 2 Report

In this manuscript, Karginov and co-authors present a very interesting study on the perturbations in the heme and siroheme biosynthesis pathways that cause the accumulation of fluorescent free base porphyrins in Ogataea yeasts. The manuscript is very well written, introducing new and sound data. There are some minor issues that the authors need to address before the manuscript can be accepted for publication.

1) Line 52: Please provide the full name of the two yeast strains utilized, as this is the first time they appear in the manuscript, apart from the abstract.

2) Section Materials and methods must be divided into sub-sections, e. g., Strains, media, and growth conditions; Plasmids; Construction of genomic library; Extraction of soluble porphyrins, etc.

3) Lines 71-73: The provenance of the strains must be clearly specified: were they obtained by academic exchange, purchased, etc.? 

4) Lines 110-111: Please specify the source of LEDs utilized.

5) Line 139: the UV mutagenesis procedure is not described in Materials and methods section.

6) Figure 4, legend: Abbreviate excitation and emission as in the diagram presented. 

Author Response

“1) Line 52: Please provide the full name of the two yeast strains utilized, as this is the first time they appear in the manuscript, apart from the abstract.”

Response 1: The full names of the yeast species have been provided.

“2) Section Materials and methods must be divided into sub-sections, e. g., Strains, media, and growth conditions; Plasmids; Construction of genomic library; Extraction of soluble porphyrins, etc.”

Response 2: The section has been divided into subsections.

“3) Lines 71-73: The provenance of the strains must be clearly specified: were they obtained by academic exchange, purchased, etc.?”

Response 3: This information has been included.

“4) Lines 110-111: Please specify the source of LEDs utilized.”

Response 4: Unfortunately the LED was purchased from a small company, which does not exist anymore. The LED does not have any markings indicating its manufacturer. Instead, we described the LED characteristics and included the name of the spectrometer, which was used to determine these characteristics, into this version of the manuscript. Any LED with similar characteristics should provide identical results.  

“5) Line 139: the UV mutagenesis procedure is not described in Materials and methods section.”

Response 5: A description of the procedure has been added.

“6) Figure 4, legend: Abbreviate excitation and emission as in the diagram presented.”

Response 6: The abbreviations in the legend and diagram have been modified to be identical.